# Prevalence of Antibiotic Resistance and Virulence Genes in *Escherichia coli* Carried by Migratory Birds on the Inner Mongolia Plateau of Northern China from 2018 to 2023

**DOI:** 10.3390/microorganisms12061076

**Published:** 2024-05-26

**Authors:** Danhong Wang, Xue Ji, Bowen Jiang, Yue Yuan, Bing Liang, Shiwen Sun, Lingwei Zhu, Jun Liu, Xuejun Guo, Yuhe Yin, Yang Sun

**Affiliations:** 1School of Chemistry and Life Science, Changchun University of Technology, Changchun 130012, China; w13134456587@163.com; 2Changchun Veterinary Research Institute, Chinese Academy of Agricultural Sciences, Changchun 130121, China; ji_xuecn@aliyun.com (X.J.); jiangbowen17@126.com (B.J.); 910449055@foxmail.com (Y.Y.); liangbing0427@163.com (B.L.); 434227760@foxmail.com (S.S.); lingweiz@126.com (L.Z.); liubio@126.com (J.L.); xuejung2021@163.com (X.G.); 3Key Laboratory of Jilin Province for Zoonosis Prevention and Control, Changchun 130121, China

**Keywords:** migratory birds, *Escherichia coli*, multidrug resistance, virulence genes, MDR transmission, East Asian–Australasian Flyway

## Abstract

(1) Background: Antibiotic resistance in bacteria is an urgent global threat to public health. Migratory birds can acquire antibiotic-resistant and pathogenic bacteria from the environment or through contact with each other and spread them over long distances. The objectives of this study were to explore the relationship between migratory birds and the transmission of drug-resistant pathogenic *Escherichia coli*. (2) Methods: Faeces and swab samples from migratory birds were collected for isolating *E. coli* on the Inner Mongolia Plateau of northern China from 2018 to 2023. The resistant phenotypes and spectra of isolates were determined using a BD Phoenix 100 System. Conjugation assays were performed on extended-spectrum β-lactamase (ESBL)-producing strains, and the genomes of multidrug-resistant (MDR) and ESBL-producing isolates were sequenced and analysed. (3) Results: Overall, 179 isolates were antibiotic-resistant, with 49.7% MDR and 14.0% ESBL. Plasmids were successfully transferred from 32% of ESBL-producing strains. Genome sequencing analysis of 91 MDR *E. coli* strains identified 57 acquired resistance genes of 13 classes, and extraintestinal pathogenic *E. coli* and avian pathogenic *E. coli* accounted for 26.4% and 9.9%, respectively. There were 52 serotypes and 54 sequence types (STs), including ST48 (4.4%), ST69 (4.4%), ST131 (2.2%) and ST10 (2.2%). The international high-risk clonal strains ST131 and ST10 primarily carried *bla*_CTX-M-27_ and *bla*_TEM-176_. (4) Conclusions: There is a high prevalence of multidrug-resistant virulent *E. coli* in migratory birds on the Inner Mongolian Plateau. This indicates a risk of intercontinental transmission from migratory birds to livestock and humans.

## 1. Introduction

*Escherichia coli* (*E. coli*) is a bacterium that resides in the gut of humans and animals in a harmless state. However, it is also a versatile pathogen that is commonly associated with intestinal and extraintestinal infections, as well as antimicrobial resistance [1]. Intestinal pathogenic *E. coli* (InPEC) strains include enterotoxigenic *E. coli*, enterohemorrhagic *E. coli*, enteroinvasive *E. coli* and enteropathogenic *E. coli* [2,3]. In addition, extraintestinal pathogenic *Escherichia coli* (ExPEC) strains are receiving increasing attention [4]. These strains are often responsible for healthcare-associated infections and prone to multidrug resistance. The use of antibiotics can significantly increase the success rate of curing bacterial infections, but their misuse has resulted in the development of new antimicrobial resistance (MDR) strains including multidrug-resistant (MDR), extensively drug-resistant (XDR) and pandrug-resistant (PDR) bacteria, all of which pose a serious public health risk [5,6].

*E. coli* is an excellent bacterial model to study the transmission routes of the above-mentioned bacterial antibiotic resistance modes because cells possess a range of antibiotic resistance modes [7,8]. Many studies have shown that *E. coli* can spread at the wildlife–livestock–human interface [9,10]. The East Asia–Australasia Flyway (EAAF) is one of the eight major bird migratory routes in the world, spanning 22 countries and involving >50 million birds from at least 250 different populations [11]. Various wetlands on the Inner Mongolia Plateau in China are important ecosystems and habitats for migratory birds flying along the EAAF migration routes [12,13].

The Inner Mongolia Plateau is located in the intersection of the EAAF and the local migratory routes of some local birds. This geographic location is very important as a breeding and transit site [14]. Many types of migrant birds inhabit this region, including wintering birds such as Anatidaes, as well as migrating birds including Larus relictus [15,16,17]. Different types of migratory birds can acquire antibiotic-resistant bacteria and pathogenic bacteria from the environment and through contact with each other, and spread them over long distances [18,19]. We previously demonstrated that the resistance level of *E. coli* in samples from migratory birds was consistent with the environment, which confirmed, at least to a certain extent, that migratory birds are potential transmitters of drug-resistant bacteria [20]. Therefore, continued studies are warranted to investigate the transmission of drug-resistant bacteria by migratory birds from temporal and spatial dimensions. To strengthen the protection of wildlife and understand the transmission risks of drug-resistant bacteria, it is of great significance to carry out antibiotic resistance and pathogen monitoring of *E. coli* carried by migratory birds in the northern wetland protected areas of China.

## 2. Materials and Methods

### 2.1. Sample Collection

Faecal and swab samples (n = 4422) were collected from migratory birds on the Inner Mongolia Plateau of northern China from 2018 to 2023. Information for the samples is shown in Table 1. Sampling procedures were approved by the Experimental Animal Welfare and Ethics Committee of Changchun Veterinary Research Institute, Chinese Academy of Agricultural Sciences. No anaesthesia, euthanasia or animal sacrifice were performed in this study. Sampling locations were in the Ningxia Hui Autonomous Region and the Inner Mongolia Autonomous Region near the northern border of China, both within the EAAF path (Figure 1). Anal and pharyngeal swabs were collected under the supervision of the Wildlife Source and Disease Inspection Station, Shenyang, China.

### 2.2. Isolation and Identification of E. coli

Each sample was placed in a 1.5 mL centrifuge tube with 1 mL physiological saline. After vortexing, 20 µL of suspension was placed on a MacConkey agar plate (Qingdao Hope Bio-Technology Co., Ltd., Qingdao, China) and cultured for 16–18 h at 37 °C. Suspected single colonies of *E. coli* were selected and purified twice (one strain per sample). Specific *E. coli* 16S rDNA primers [21] were used to identify the isolates. PCR was carried out in ETC821 (Eastwin Life Sciences Inc., Beijing, China). Reactions contained 12.5 μL 2 × taq enzyme (Cowin Biosciences Inc., Beijing, China), 0.5 μL upstream primer, 0.5 μL downstream primer, 10.5 μL ddH_2_O and 1 μL template DNA. The thermal conditions are listed in Appendix A. Positive strains were further determined using a BD Phoenix Automated Identification and Susceptibility Testing System (Becton, Dickinson and Company, Franklin Lakes, NJ, USA). 

### 2.3. Antibiotic Susceptibility Testing

Twenty-one antibiotics were used to identify resistance phenotypes of *E. coli* isolates. Intestinal pathogenic *E. coli* was used (InPEC NMIC/ID 4 panel of a BD Phoenix Automated Identification and Susceptibility Testing System). The antibiotics included the aminoglycosides amikacin (AMK) and gentamicin (GEN); the carbapenems imipenem (IPM) and meropenem (MEM); the cephalosporins cefazolin (CZO), ceftazidime (CAZ), cefotaxime (CTX) and cefepime (FEP); the monobactams aztreonam (ATM), ampicillin (AMP) and piperacillin (PIP); the penicillins + β-lactamase inhibitors amoxicillin-clavulanate (AMC) and ampicillin-sulbactam (SAM); the antipseudomonal penicillins + β-lactamase inhibitors piperacillin-tazobactam (PEF); the polymyxin colistin (COZ); the sulfonamide trimethoprim-sulfamethoxazole (STX); the phenicol chloramphenicol (CHL); the fluoroquinolones ciprofloxacin (CIP), levofloxacin (LVX) and moxifloxacin (MXF); and tetracycline (TET). The ESBL phenotype was identified by the automatic drug susceptibility system. For potential ESBL-producing strains, the corresponding ESBL resistance markers are shown in the report. The MDR phenotype was defined as resistance to ≥3 antimicrobial classes [22].

### 2.4. Minimum Inhibitory Concentration (MIC)

MICs were determined using the microbroth dilution method, performed according to the Clinical and Laboratory Standards Institute (CLSI) guidelines (CLSI, 2022: M100-S25). Sixteen antibiotics (Shanghai Yuanye Bio-Technology Co., Ltd., Shanghai, China) belonging to ten antibiotic classes were tested: aminoglycosides (GEN), 1st generation cephalosporins (CZO), 3rd and 4th generation cephalosporins (CAZ, CTX and FEP), monobactams (ATM), penicillins (AMP and PIP), penicillins + β-lactamase inhibitors (AMC and SAM), sulfonamides (SXT), phenicols (CHL), fluoroquinolones (CIP, LVX and MXF) and tetracyclines (TET). The quality control, carried out by testing the reference strains of *E. coli* ATCC 25922, confirmed compliance with the manufacturer’s declared antibiotic concentration ranges. The MIC values obtained in this study of antibiotic susceptibility were determined according to the criteria of the CLSI [23].

### 2.5. Conjugation Experiments

Plasmid transferability was determined using rifampicin-resistant *E. coli* EC600 as the recipient. ESBL-producing *E. coli* isolates and *E. coli* EC600 were cultured for 12 h at 37 °C, then mixed at a ratio of 1:1. Samples (0.1 mL) of mixed culture were filtered through nitrocellulose filters. Then, the filter membrane was placed on an LB agar plate and incubated for 12 h at 37 °C. Transconjugants were selected on LB agar plates containing rifampicin (800 μg/mL) (Solarbio, Beijing, China) and cefotaxime (4 μg/mL) (China National Institutes for Drug Control, Beijing, China). They were further identified to confirm the ESBL and antibiotic resistance phenotypes.

### 2.6. Whole-Genome Sequencing and Analysis

MDR and ESBL-producing isolates were selected for sequencing. Genomic DNA of isolates was extracted using a bacterial DNA extraction kit (Omega Bio-Tek Company, Norcross, GA, USA), subjected to agarose gel electrophoresis and quantified using a Qubit 2.0 Fluorimeter (Thermo Scientific, Waltham, MA, USA). Whole-genome sequencing was performed using an Illumina NovaSeq PE150 platform at Beijing Novogene Bioinformatics Technology Co., Ltd. (Beijing, China). Predictions of virulence genes and antimicrobial resistance genes (ARGs) were based on the Virulence Factors of Pathogenic Bacteria (VFDB) database (http://www.mgc.ac.cn/cgi-bin/VFs) (accessed on 27 July 2023) and the Comprehensive Antibiotic Research Database (CARD) database (https://card.mcmaster.ca/) (accessed on 27 July 2023). Comparison of 2nd generation sequencing results for resistance genes (https://cge.cbs.dtu.dk/services/KmerResistance/) (accessed on 5 August 2023), virulence genes (https://cge.cbs.dtu.dk/services/VirulenceFinder/) (accessed on 5 August 2023), serotypes (https://cge.cbs.dtu.dk/services/SerotypeFinder/) (accessed on 5 August 2023), plasmid types (https://cge.cbs.dtu.dk/services/PlasmidFinder/) (accessed on 5 August 2023) and MLST typing (https://cge.cbs.dtu.dk/services/MLST/) (accessed on 5 August 2023) was performed using the CGE website (https://cge.cbs.dtu.dk/services/) (accessed on 5 August 2023).

### 2.7. PCR Identification of ARGs

The major resistance genes carried by MDR and ESBL strains based on whole-genome sequencing results are listed in Appendix A. And transconjugants of ESBL-producing isolates were identified by PCR assay. Ten representative ARGs were assessed including *bla*_CTX-M_, *bla*_TEM-1_, *tet*(A), *tet*(B), *tet*(M), *sul1*, *sul2*, *sul3*, *floR* and *cmlA*. Details are listed in Appendix A. PCR was carried out with ETC821 (Eastwin Life Sciences Inc., Beijing, China). Reactions contained 12.5 μL 2 × taq enzyme (Cowin Biosciences Inc., Beijing, China), 0.5 μL upstream primer, 0.5 μL downstream primer, 10.5 μL ddH_2_O and 1 μL template DNA.

### 2.8. Phylogenetic Analysis

For phylogenetic analysis, genome sequences were aligned and analysed. Phylogenetic trees were constructed by Realphy [24], Bowtie2 [25] and RAxML [26] based on single-nucleotide polymorphism (SNP) sites using the maximum likelihood method. In addition to the whole-genome sequences of 91 strains in this study (all MDR and ESBL-producing isolates), 40 whole-genome sequences of *E. coli* were downloaded from the NCBI database. Most of these strains originate from wild animals. Some reference strains were isolated from local domesticated animals (Appendix A).

## 3. Results

### 3.1. Isolation and Identification of E. coli

A total of 1327 (30.0%) *E. coli* strains were isolated from 4422 samples. The isolation rates of *E. coli* from Ningxia and Inner Mongolia were 79.5% (1055/1327) and 20.5% (272/1327), respectively; 58.6% (777/1327) in spring and 41.4% (550/1327) in autumn. The isolation rates are listed in Table 2. 

### 3.2. Antibiotic Susceptibility Testing 

Overall, 13.5% (179/1327) of isolates exhibited resistance to at least one antibiotic. The resistance rates of isolates from Ningxia and Inner Mongolia were 12.3% (130/1055) and 18.0% (49/272), respectively, with 14.0% (109/777) in spring and 12.7% (70/550) in autumn. Details for every year are shown in Figure 2. 

A total of 179 antibiotic resistance strains were resistant to 17 antibiotics. The resistance rates for tetracycline, ampicillin, piperacillin, trimethoprim/sulfamethoxazole, chloramphenicol and cefazolin were 11.1% (147/1327), 9.1% (121/1327), 7.9% (105/1327) 5.1% (69/1327), 5.1% (68/1327) and 2.1% (28/1327), respectively; the resistance rates for ciprofloxacin, moxifloxacin, cefotaxime, levofloxacin, cefepime, aztreonam, ampicillin/sulbactam, gentamicin, amoxicillin/clavulanate, colistin and ceftazidime ranged from 0.1% (1/1327) to 1.8% (25/1327). The isolates were highly or moderately sensitive to imipenem, meropenem and piperacillin/tazobactam, and 14.0% (25/179) of isolates produced ESBLs. Details are shown in Figure 3.

Overall, 49.7% (89/179) of antibiotic resistance isolates were MDR, and the most frequent combination of antibiotic resistance was ampicillin–piperacillin–trimethoprim/sulfamethoxazole–chloramphenicol–tetracycline (13 isolates). More than 21 (23.6%) isolates showed resistance to six or more antimicrobial classes. Details are shown in Appendix A.

### 3.3. Conjugation Experiments

Overall, 32% (8/25) of the ESBL-producing isolates successfully transferred their plasmids. Seven transconjugants produced new antibiotic resistance phenotypes, including CAZ (MIC 16–64 μg/mL), ATM (MIC 4–16 μg/mL), MXF (MIC 32 μg/mL) and CIP (MIC 16 μg/mL), and the MIC values of the six antibiotics increased 2–4 times in the transconjugants. The same antibiotic resistance phenotype showed different MIC values in the transconjugants. The antibiotic resistance phenotypes and MIC values of the isolates and their transconjugants are listed in Table 3.

### 3.4. Bioinformatics Analysis

#### 3.4.1. Analysis and Verification of ARGs

The genome analysis of 91 strains identified 64 types of acquired resistance genes (similarity ≥ 90%) of 14 classes, including β-lactam resistance genes (*bla*_CTX-M_, *bla*_TEM-1_), tetracycline resistance genes (*tet* (A), *tet* (B), *tet* (M)), sulfonamide resistance genes (*sul1*, *sul2*, *sul3*), chloramphenicol resistance genes (*FloR*, *cmlA*) and aminoglycoside resistance genes (*aph(3″)-Ib*, *aadA5*, *aadA1*). Overall, 52.7% (48/91) of isolates carried the *sul2* gene, alone or in combination with *sul1* or *sul3* endowing sulfonamide resistance. Details are shown in Appendix A. All 25 ESBL-producing isolates carried the *bla*_CTX−M_ genes (*bla*_CTX-M-1_, *bla*_CTX-M-14_, *bla*_CTX-M-15_, *bla*_CTX-M-27_, *bla*_CTX-M-55_), of which *bla_CTX−M-55_* was the most dominant genotype (32.0%, 8/25). Overall, 36.0% (9/25) of ESBL-producing isolates harboured both *bla*_CTX−M_ and *bla_TEM−1_* genes, while *bla*_CTX-M-65_ and *bl*_aCMY-2_ were found in some ESBL-producing *E. coli* strains.

Confirmation of ARGs in MDR isolates by PCR assay showed that *bla_CTX-M_* was 100% (25/25), *bla_TEM-_*_1_ was 90.5% (57/63), *sul1* was 60% (9/15), *sul2* was 97.9% (47/48), *sul3* was 100% (14/14), *floR* was 92.2% (47/51), *cmlA* was 75% (12/16), tet(A) was 95.1% (77/81), *tet*(B) was 66.7 (2/3) and *tet*(M) was 100% (5/5).

According to the whole-genome sequencing and PCR results, in eight transconjugants *bla_CTX-M_*, *sul1*, *sul2*, *sul3*, *floR*, *cmlA*, *tet*(A), *tet*(B) and *tet*(M) were carried in the same way as in the donor strains. However, five of the eight donor strains carried *bla*_TEM-1_, but none of the transconjugants carried *bla*_TEM-1_.

#### 3.4.2. Analysis of Plasmid Types

Seventeen replicon types of plasmids were detected of which IncF was the most common. In order of abundance the diverse replicon types were IncFIB, IncFIC, IncFII, IncY, p0111, IncX1, IncFIA, IncI1-I, Col156, IncB/O/K/Z, IncQ1, IncHI2, IncHI2A, IncX4, Col (BS512), IncI and IncN. The top four types (IncFIB, IncFIC, IncFII, IncY) were the same in both Ningxia and Inner Mongolia. Overall, 33.0% (30/91) of isolates carried more than one plasmid and 4.4% (4/91) of isolates were not detected. The plasmid replicon types of each isolate are listed in Appendix A.

#### 3.4.3. Analysis of Virulence-Associated Genes (VAGs)

The VAG analysis of 91 sequenced strains showed the presence of typical virulence genes of pathogenic *E. coli*, such as EAST1 heat-stabilising toxin *astA* in 15.4% (14/91); cytotoxic necrotoxicity factor *cnf1* in 2.2% (2/91); and the iron carrier receptors *ireA*, *iroN*, *irp2* and fyuA and enterotoxin *sen* in 5.5% (5/91), 33.0% (30/91), 23.1% (21/91), 22.0% (20/91) and 1.1% (1/91), respectively. 

Isolates were classified operationally as ExPEC if they contained at least two of the five ExPEC-defining virulence genes (*papA* and/or *papC*, *sfa/focDE*, *afa/draBC*, *kpsM II* and *iutA*) [27,28]. Additionally, isolates were classified operationally as APEC if they contained four or more VAGs (*papC*, *iucD*, *irp2*, *tsh*, *vat*, *astA*, *iss*, *cva* and *cvi*) [27].

Of the 91 MDR and ESBL isolates, 26.4% (24/91) were qualified molecularly as ExPEC, and of these, 29.2% (7/24) were further qualified as APEC. Another two isolates qualified molecularly as APEC but not ExPEC, yielding 9.9% (9/91) UPEC isolates.

Among the 24 ExPEC strains, 70.8% (17/24) were from Ningxia and 29.2% (7/24) were from Inner Mongolia. Of the nine APEC strains, 88.9% (8/9) were from Ningxia and 11.1% (1/9) were from Inner Mongolia.

One APEC strain had all nine virulence genes, two had five virulence genes and six had four virulence genes. *irp2* and *iss* were detected in all nine APEC strains. The virulence genes *tsh*, *vat*, *astA* and *cva* were detected in five, five, three and seven APEC strains, respectively, while *cva* and *iucD* were not detected. The virulence genes of each isolate are shown in Appendix A.

#### 3.4.4. Analysis of Phylogenetic Relationships

Fifty-four different STs were observed with a relatively prevalent type of ST2448 (n = 5), followed by ST48 (n = 4), ST69 (n = 4) and ST351 (n = 4). Internationally and importantly prevalent types were also found (Appendix A), including ST10 (n = 2) and ST131 (n = 2). 

The phylogeny of the 91 *E. coli* isolates and 40 reference *E. coli* strains was analysed by constructing a neighbour-joining phylogenetic tree based on SNP sites (Figure 4). The tree revealed four major phylogroups containing some smaller clusters. Remarkably, all strains from other countries (2010–2019) were grouped into the first two big clusters containing 23.1% (21/91) of isolates (2021–2023) in this study. Meanwhile, 49 isolates and six reference strains in which five strains were from local domesticated animals monopolised the third branch. Strains isolated from *L. relictus* had unique ST types and were in relatively independent phylogenetic branches. Most of the isolates clustered together in accordance with the same ST types.

## 4. Discussion

Migratory birds are known to carry antibiotic-resistant *E. coli* [29]. Waterfowl can be considered sentinels of AMR in the environment [30]. *E. coli* is an excellent indicator species for studying the spread of AMR at the wildlife–livestock–human interface. In previous studies, MDR *E. coli* were detected in populations of migratory birds from various parts of China [31,32]. However, there remains a lack of in-depth research on the drug-resistant bacteria carried by migratory birds from the dimensions of time and space. In this study, faeces and swab samples from migratory birds on the Inner Mongolian Plateau of China were collected from 2018 to 2023. Continuous monitoring was carried out for drug-resistant *E. coli*, and the drug resistance spectra of isolates were determined. ESBL strains underwent coupling experiments to evaluate transmission ability, coupled with conjugation assays. To better understand the genetic evolution of isolates and assess the risk of spreading ARGs and VRGs, the genomes of MDR *E. coli* were sequenced and analysed. 

Overall, the resistance rate of *E. coli* isolates from migratory birds in this study was 13.5%, lower than reported in other studies on *E. coli* derived from migratory birds [33]. The sampling area in this study is located northwest of the Aihui–Tengchong line, which demarcates both population and urban development. Carbapenem-resistant and colistin-resistant *E. coli* have been isolated from faecal samples of migratory birds southeast of the Aihui–Tengchong line, including Shanghai, Jiangsu and Zhejiang [34]. In this study, carbapenem-resistant *E. coli* was not isolated. Although colistin-resistant *E. coli* was included among the isolated strains, the proportion was very low. The isolation rate of MDR *E. coli* in this study is much lower than that in the southeastern region of the Aihui–Tengchong line [35]. These results to some extent reflect the impact of human activities on wildlife. The drug resistance rate of *E. coli* isolates from Inner Mongolia was higher than that of isolates from Ningxia. A large number of antibiotics were used early in the development of Inner Mongolia’s livestock industry. From a timeline perspective, the overall trend of antibiotic resistance rates in Ningxia and Inner Mongolia is decreasing year by year, which may be related to China’s strict policy of restricting the use of antibiotics in the animal husbandry industry since 2020. This impact might have a lag period, and other influencing factors such as climate cannot be ignored [36].

From the perspective of drug resistance phenotypes, the *E. coli* isolates had the highest resistance rate to tetracycline, followed by penicillin. This is directly related to the widespread use of tetracyclines in the early stages of animal husbandry in China [37]. In most cases, animal-derived *E. coli* typically have a high resistance rate to early antibiotics, including tetracyclines and sulfonamides. The tetracycline resistance gene *tet*(A) (89.0%) detected in this study is the dominant gene involved, while others include the tetracycline resistance genes *tet*(B) (3.3%) and *tet*(M) (5.5%). The active efflux genes *tet*(A) and *tet*(B) and the ribosomal protective gene *tet*(M) can be transferred between bacteria through plasmids and transposons, leading to widespread tetracycline resistance [38,39]. 

The worldwide spread of ESBL-producing bacteria, particularly *E. coli*, is a critical concern for the development of therapies against MDR bacteria. Various factors, such as environmental sources, food animals and international travel, accelerate global ESBL spread [40]. Migratory birds, due to their intercontinental migration behaviour and overlapping with human habitats, influence the global spread of ESBL. It is worth noting that 25 out of all isolates were ESBL-producing *E. coli* in our study, and all carried *bla*_CTX−M_ genes. Some studies have reported a global distribution of *bla*_CTX-M_ [41], and the origin of *bla*_CTX-M_-type ESBLs is closely related to humans [42]. In a study in Pakistan, whole-genome sequencing showed that *bla*_CTX-M-15_ is the major ESBL [43], consistent with some previous studies in migratory birds from Bangladesh [44], Germany [45] and North America [46]. However, *bla*_CTX-M-55_ and *bla*_CTX-M-14_ were the most prevalent ESBL genotypes in our study. Recent surveillance studies showed that CTX-M-27 is emerging in certain parts of the world, especially Japan and Europe [47]. We found that three ESBL-*E. coli* from migratory birds carried *bla*_CTX-M-27_ genes, and one belonged to a global high-risk clone called ST131. 

The results of conjugation experiments showed that ESBLs of eight strains were successfully transferred, and the plasmid type was mainly IncF. Some interesting phenomena were found in the conjugation experiments; four conjugates increased the resistance phenotype of ceftazidime compared to donors, with MIC values of 32, 16, 64 and 16 μg/mL. This may be related to the expression of a multidrug efflux pump [48]. Another possible explanation is that the expression of exogenous antibiotic resistance genes carried by plasmids was inhibited in the donors. There were also two ESBL-*E. coli* donors that did not have a quinolone-resistant phenotype but carried the quinolone resistance gene qnrS1. After conjugation, the transconjugants also obtained qnrS1 and showed a quinolone-resistant phenotype. This indicates that the regulation of drug resistance gene expression was closely related to chromosomes. 

In addition to some typical VRGs carried by intestinal pathogenic *E. coli* (InPEC), many VRGs of extraintestinal pathogenic *E. coli* (ExPEC) were identified in this study. ExPEC are classified into the following different types based on their infected host and clinical symptoms: Neonatal Meningitis *E. coli* (NMEC), Uropathogenic *E. coli* (UPEC), Avian Pathogenic *E. coli* (APEC) and Septic *E. coli* (SEPEC) [3]. Based on studies on genetic similarity and pathogenic similarity, APEC iss closely related to human UPEC and SEPEC [49]. This suggests that APEC may be a reservoir for human ExPEC virulence and resistance genes, and virulence and resistance genes could be transferred to humans through animals. According to the determination criteria of ExPEC and APEC [27,28], 24 *E. coli* isolates were identified as ExPEC and 9 *E. coli* isolates were identified as APEC. Additionally, 20 *E. coli* isolates belonged to the potentially highly virulent APEC, and 7 of the 9 APEC isolates belonged to ExPEC. One (210326G31) of the other two APEC strains was an InPEC-APEC heterozygous pathogen. The other strain (21ZN102dc) carried a VRG of UPEC and a VRG of ExPEC, but it was not sufficient to determine whether it was UPEC or ExPEC. 

These isolates carrying VRGs were relatively dispersed on the phylogenetic tree but combined with multi-locus sequence typing (MLST) and serotype analysis, clues of cross-continental transmission of MDR high-risk clones have been discovered. In the present study, ExPEC 22ZNG-196dc isolated from Ningxia in 2022 was evolutionarily related to *E. coli* 63CIPA isolated from birds in Australia in 2017 [50], both of which belonged to the ST131 O25:H4 international MDR high-risk clone. Australia is also located in the EAAF. An APEC 23NH-39EC from Ordos in Inner Mongolia in 2023 was evolutionarily related to two *E. coli* strains from wild Arctic caribou in 2011 [51] and all belong to ST131 O25:H4. The Arctic is also located along the same migration route. ExPEC 21ZN33dc isolated from Ningxia in 2021 was evolutionarily related to *E. coli* D47_7 from sewage in South Africa in 2017 [52], and both belong to ST69. ExPEC 22NPY-117 isolated from Ningxia in 2022 was evolutionarily related to *E. coli* U40_6 from South Africa in 2018 [52], but belonged to a unique sequence type (ST1188). In our study, two classical APEC serotypes (O2 and O18) were identified, both of which are present worldwide. In addition, some typical ESBL-associated sequence types were identified including ST10, ST69 and ST131. These sequence types involved in the widespread spread of resistance to antibiotics are critical in human medicine and have been reported in clinical studies in China [53,54]. This suggests that there may be cross-transmission between humans and migratory birds. Some specific sequence types (ST48, ST93, ST155, ST162, ST297, ST351, ST602, ST906, ST1844 and ST 2526) were dominant clones of local isolates, and the same sequence types were correlated with regions. They were densely distributed on the evolutionary tree, mostly in later evolutionary branches. Some sequence type isolates (ST48, ST155, ST602 and ST906) were closely related to human or domestic animal isolates [55,56]. This further proves the close relationship between human activities and epidemic strains of wildlife. 

Interestingly, most of the isolated strains of *L. relictus* were in relatively independent and late evolutionary branches, with the dominant sequence type being ST2448, but there are also two exceptions (ST 1125 and ST609). The relict gull is one of the most threatened gulls in the world [57]. Internationally, it is listed as vulnerable according to the Convention on International Trade in Endangered Species of Wild Fauna and Flora (CITES) and the Convention on the Conservation of Migratory Species of Wild Animals (CMS) Appendices I. The largest colonies in China are known as the Ordos subpopulation [58]. The dominant genera in the gut microbiota of Relict gulls are Escherichia and Shigella, and the gut harbours numerous pathogenic bacteria [59]. Members of the Ordos relict gull population migrate in an east–west direction along the East Asian continent, and the breeding grounds on the Inner Mongolian Plateau intersect with other migratory birds migrating south–north along the EAAF route. In this study, the other sampling migratory birds were mainly Anatidae. The resistance rate of *E. coli* isolates from these migratory birds was 15.2%, while the resistance rate of those from the relict gull was 18.4%. The reasons for this phenomenon may be related to genetics, lifestyle habits and food sources. Relict gulls are omnivorous animals, and they also have close contact with tourists in protected areas [57,60]. Anatidae species mainly feed on fish and shrimp in the water, have high vigilance and are accustomed to keeping a distance from humans [61]. However, in a common ecological environment, the intersection of habitats may lead to the acquisition of and variations in multi-virulence and MDR strains among different species, resulting in the wider spread of risk clones.

## 5. Conclusions

In the present study, the molecular epidemiological backgrounds of *E. coli* carried by migratory birds on Inner Mongolian Plateau wetlands were analysed in two dimensions (time and space). Important epidemiological information on the global spread of important resistance and virulence genes carried by migratory birds was acquired. The results showed that the MDR bacteria carried by migratory birds in the Inner Mongolian Plateau wetlands were closely related to the genetic background of epidemic strains in other regions along the migration route of the EAAF. Migratory birds can carry MDR bacteria for intercontinental transmission. The results also reflected the transmission patterns and characteristics of local epidemic strains of *E. coli* carried by migratory birds. This indicates a risk of transmission of pathogenic *E. coli* to humans from migratory birds.

## Figures and Tables

**Figure 1 microorganisms-12-01076-f001:**
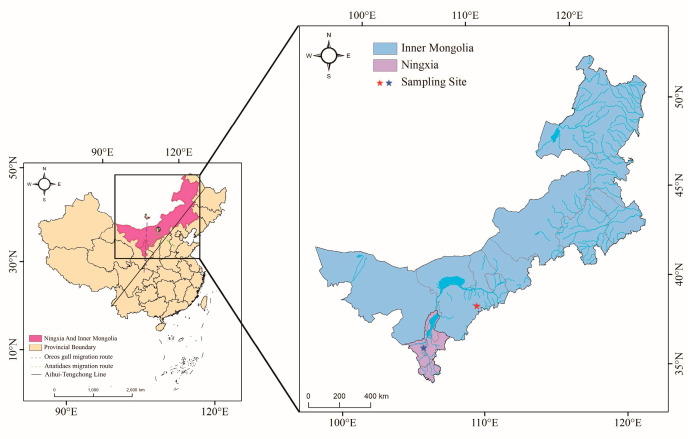
Sampling locations of migratory birds.

**Figure 2 microorganisms-12-01076-f002:**
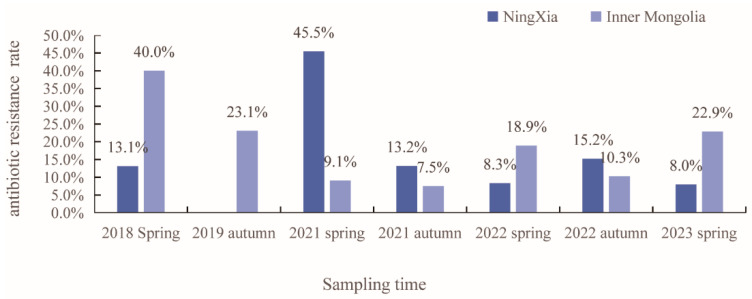
Antibiotic resistance rates of isolates from migratory birds over 5 years.

**Figure 3 microorganisms-12-01076-f003:**
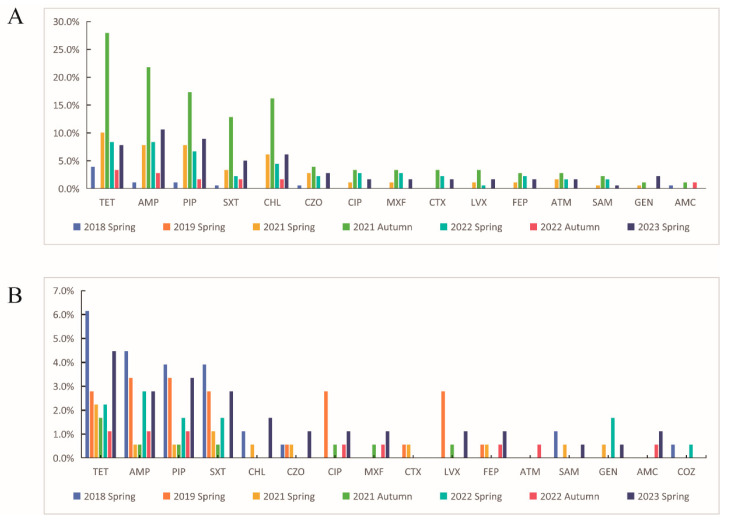
Antibiotic resistance of isolates over 5 years (2018–2023): (**A**) Ningxia province (**B**) Inner Mongolia province.

**Figure 4 microorganisms-12-01076-f004:**
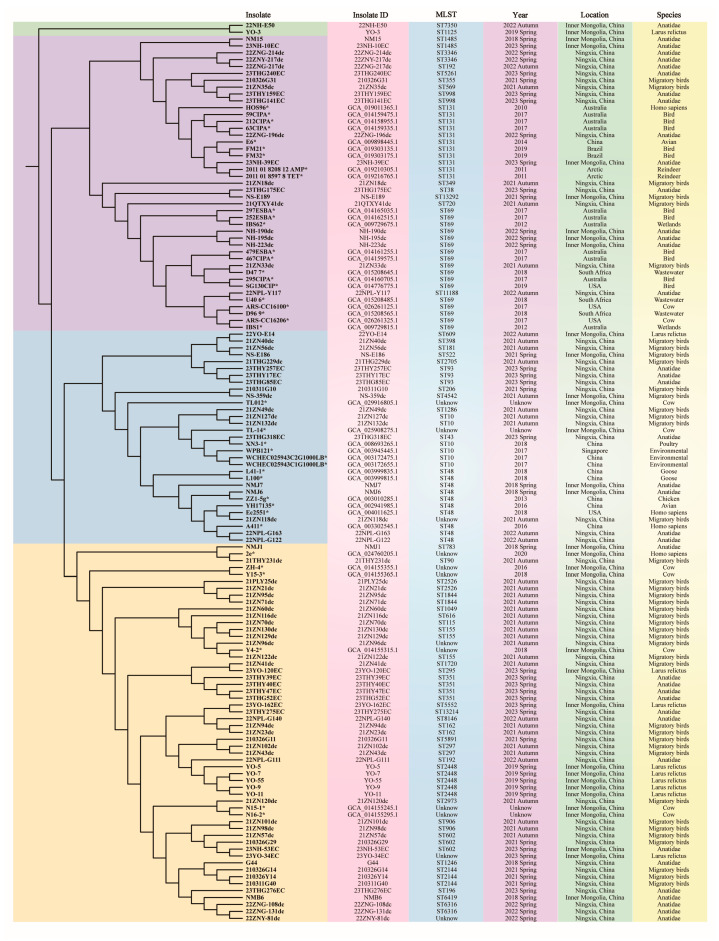
Phylogenetic tree of 91 isolates and 40 reference strains. (The “*” depicted in the image represents the NCBI reference strain.)

**Table 1 microorganisms-12-01076-t001:** Sample information.

Sampling Time	Sampling Locations	Bird spp.	Sample Type	Number of Samples
Spring 2018 [20]	Tianhu Lake, Ningxia	Anatidae	swab	214
	Qingtongxia			
	Huanghe Beach, Wuzhong City			
	Honghaizi Wetland, Inner Mongolia	Anatidae	faeces	125
		*Larus relictus*		
Spring 2019 [20]	Honghaizi Wetland, Inner Mongolia	*Larus relictus*	faeces	82
Spring 2021	Ningxia Pingluo Yellow River wetland forest farm	Anatidae	swab	292
	Kangbashi, Ordos, Inner Mongolia	Anatidae	faeces	320
		*Larus relictus*		
Autumn 2021	Tianhu Lake, Ningxia	Anatidae	swab	845
	Ningxia Pingluo Yellow River wetland forest farm			
	Kangbashi, Ordos, Inner Mongolia	*Larus relictus*	faeces	140
Spring 2022	Tianhu Lake, Ningxia	Anatidae	swab	532
	Qingtongxia			
	Honghaizi Wetland, Inner Mongolia	*Larus relictus*	faeces	360
Autumn 2022	Ningxia Pingluo Yellow River wetland forest farm	Anatidae	swab	359
	Honghaizi Wetland, Inner Mongolia	*Larus relictus*	faeces	202
Spring 2023	Tianhu Lake, Ningxia	Anatidae	swab	636
	Honghaizi Wetland, Inner Mongolia	Anatidae	faeces	315
		*Larus relictus*		

**Table 2 microorganisms-12-01076-t002:** Isolation and drug resistance rates for *E. coli* in Ningxia and Inner Mongolia.

Sampling Time	Number of Strains		Number of Samples		Isolation Rate of *E. coli* (%)		Antibiotic Resistance Rate (%)	
	Ningxia	Inner Mongolia	Ningxia	Inner Mongolia	Ningxia	Inner Mongolia	Ningxia	Inner Mongolia
Spring 2018 [20]	61	40	214	125	28.5	32.0	13.1	40.0
Spring 2019 [20]	-	26	-	82	-	31.7	-	23.1
Spring 2021	44	55	292	320	15.1	17.2	45.5	9.1
Autumn 2021	425	40	845	140	50.3	28.6	13.2	7.5
Spring 2022	228	37	532	360	42.9	10.3	8.3	18.9
Autumn 2022	46	39	359	202	12.8	19.3	15.2	10.3
Spring 2023	251	35	636	315	39.5	11.1	8.0	22.9
Total	1055	272	2878	1544	39.1	17.6	12.3	18.0

**Table 3 microorganisms-12-01076-t003:** Characteristics of donor strains and transconjugants.

Isolates	Antibiotic Resistance of Donor Strains	MIC (μg/mL)	Antibiotic Resistance of Transconjugants	MIC (μg/mL)
210311G10	GEN-CZO-CTX-FEP-ATM-AMP-PIP-SXT-CHL-CIP-LVX-MXF-TET	64-512-256-16-32-512-512-128-64-32-8-16-64	GEN-CZO-CAZ-CTX-FEP-ATM-AMP-PIP-SXT-CHL-TET	32-512-32-128-8-32-512-512-64-64-32
21ZN21dc	CZO-CTX-FEP-ATM-AMP-PIP-CHL-TET	512-512-8-32-512-512-64-128	CZO-CAZ-CTX-FEP-ATM-AMP-PIP-CHL-TET	512-32-256-16-128-512-512-32-64
21ZN33dc	CZO-CTX-FEP-AMP-PIP-SXT-TET	512-512-8-512-256-128-32	CZO-CTX-FEP-AMP-PIP-SXT-TET	512-256-16-512-512-64-32
21ZN71dc	CZO-CTX-FEP-ATM-AMP-PIP-CHL-TET	512-128-4-32-512-512-64-128	CZO-CAZ-CTX-FEP-ATM-AMP-PIP-CHL-TET	512-64-128-8-64-512-512-32-128
21ZN95dc	CZO-CTX-FEP-ATM-AMP-PIP-CHL-TET	512-256-8-32-512-512-128-128	CZO-CAZ-CTX-FEP-ATM-AMP-PIP-CHL-TET	512-16-256-16-128-512-512-128-128
23THY275EC	CZO-CTX-FEP-AMP-PIP	256-32-8-512-128	CZO-CTX-FEP-ATM-AMP-PIP-MXF	512-128-8-16-512-256-32
23YO-162EC	GEN-CZO-CTX-FEP-AMP-PIP-SXT-CHL-TET	16-512-32-4-512-256-128-64-64	GEN-CZO-AMP-PIP-SXT-CHL-TET-CIP-MXF	16-256-512-128-64-64-32-16-32
23NH-10EC	CZO-CTX-FEP-AMP-PIP-AMC-SAM-SXT-CHL-TET	512-128-16-512-512-8-16-128-64-64	CZO-CTX-FEP-ATM-AMP-PIP-SAM	256-128-8-4-512-512-16

Note: gentamicin (GEN), cefazolin (CZO), cefotaxime (CTX), cefepime (FEP), aztreonam (ATM), ampicillin (AMP), piperacillin (PIP), trimethoprim-sulfamethoxazole (SXT), chloramphenicol (CHL), ciprofloxacin (CIP), levofloxacin (LVX), moxifloxacin (MFX), tetracycline (TET), ceftazidime (CAZ).

## Data Availability

For samples from 2018–2019, whole-genome sequencing information of some isolated strains was collected and described as part of a prior investigation (unpublished). All sequences are publicly available in the sequence read archive under accession number PRJNA1029772, though information was included for comparative analyses as part of this project. All other genomic information is presented here for the first time. All sequences are publicly available in the sequence read archive under accession number PRJNA1061121.

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
