# Peer review of "Prevalence of Antibiotic Resistance and Virulence Genes in Escherichia coli Carried by Migratory Birds on the Inner Mongolia Plateau of Northern China from 2018 to 2023"

_microorganisms, 2024, doi:10.3390/microorganisms12061076_

Round 1
Reviewer 1 Report
Comments and Suggestions for Authors
The article is very interesting. Antibiotic resistance and its transmission by wildlife is very interesting. The authors have described the methods and results very well. My comments are minor and concern minor modifications in the manuscript.
Line 35-36 This sentence is unclear, if possible please rephrase
Line 43- please change to pandrug resistant and give the abbreviation (PDR)
Isolation and identyfication There is no information about PCR reaction, please add this (thermal coditions, names of reagents and equipment)
MIC why were less antibiotics used in the MIC?
PCR identyfication of ARGs There is no information about PCR equipment
Bioinformatic analysis please add information on why only 91 strains were used
gene names should be in italics throughout the article
Reviewer 2 Report
Comments and Suggestions for Authors
The main aim of this study was to investigate the role of migratory birds collected in the Inner Mongolia Plateau of northern China from 2018 to 2023 of carrying drug-resistant pathogenic Escherichia coli.
Several aspects to be considered for improving the manuscript include:
The authors must explain the reason of not including any quality control strain in the antibiotic susceptibility testing assays. Otherwise, breakpoints used to categorize isolates as antimicrobial resistant must be described, or a proper reference of a used guideline must be included.
Figure 1 must be discarded considering that shown information was already included in Table 2.
Figures 2 and 3 were not correctly located in the manuscript. Figure 2 must be located before Figure 3.
Figure 4 was not included in the manuscript despite was cited in text (line 259).
Size and quality of Figure inserted in page 10 must be improved. Otherwise, Figure included in page 10 has two different legends (line 269).
The phenotypic procedure used to detect the ESBL production must be properly described.
A reference must be included at the end of paragraph between lines 308-310.
Paragraph in lines 222-223 must be changed.
The authors must include the genomic data confirming the presence of the ARGs detected (percentages of identity), most probably in a supplementary table.
The authors must explain the acquisition of several antimicrobial resistances in the transconjugants despite these resistances were absent in the donors (CAZ for 21ZN71dc and 21ZN95dc isolates), (ATM for 23THY275E and 23NH-10EC isolates) and (MXF for 23THY275E isolate), as shown in Table 3.
Minor remarks:
Line 18: “for insolating” must be changed to read “for isolating”
Line 221: “et(B)” must be changed to read “tet(B)”
Lines 306-310: “tet” term must be written in italics.
Comments on the Quality of English LanguageNo further comments
